# Real-World Patient Characteristics and Treatment Patterns of Naldemedine for the Treatment of Opioid-Induced Constipation in Patients with Cancer: A Multicenter Retrospective Chart Review Study

**DOI:** 10.3390/medicina57111233

**Published:** 2021-11-11

**Authors:** Eriko Hiruta, Yukiyoshi Fujita, Hisao Imai, Takashi Masuno, Shigeki Yamazaki, Hajime Tanaka, Teruhiko Kamiya, Masako Ito, Satoshi Takei, Masato Matsuura, Hiromi Nishiba, Junnosuke Mogi, Mie Kotake, Shiro Koizuka, Koichi Minato

**Affiliations:** 1Division of Pharmacy, Gunma Prefectural Cancer Center, Ota 373-8550, Gunma, Japan; e-hiruta@gunma-cc.jp; 2Division of Respiratory Medicine, Gunma Prefectural Cancer Center, Ota 373-8550, Gunma, Japan; m06701014@gunma-u.ac.jp (H.I.); m10702012@gunma-u.ac.jp (M.K.); kminato@gunma-cc.jp (K.M.); 3Department of Respiratory Medicine, Comprehensive Cancer Center, International Medical Center, Saitama Medical University, Hidaka 350-1298, Saitama, Japan; 4Division of Pharmacy, Fujioka General Hospital, Fujioka 375-8503, Gunma, Japan; ta-masuno@fujioka-hosp.or.jp; 5Division of Pharmacy, Kiryu Kosei General Hospital, Kiryu 376-0024, Gunma, Japan; yakuzaibu1@kosei-hospital.kiryu.gunma.jp; 6Division of Pharmacy, Haramachi Red Cross Hospital, Agatsuma 377-0882, Gunma, Japan; yakuzai@haramachi.jrc.or.jp; 7Division of Pharmacy, Tatebayashi Kosei General Hospital, Tatebayashi 374-8533, Gunma, Japan; t-kamiya@tatebayashikoseibyoin.jp; 8Division of Pharmacy, Ota Memorial Hospital, Ota 373-8585, Gunma, Japan; m.itou.03464@ota-hosp.or.jp; 9Division of Pharmacy, Tone Central Hospital, Numata 378-0012, Gunma, Japan; yak-tone03@tonehoken.or.jp; 10Division of Pharmacy, Gunma Saiseikai Maebashi Hospital, Maebashi 371-0821, Gunma, Japan; ma-matsuura@maebashi.saiseikai.or.jp; 11Division of Pharmacy, JCHO Gunma Chuo Hospital, Maebashi 371-0025, Gunma, Japan; nishiba-hiromi@gunma.jcho.go.jp; 12Division of Pharmacy, Hidaka Hospital, Takasaki 370-0001, Gunma, Japan; yakuzai2147@hidaka-kai.com; 13Division of Palliative Care, Gunma Prefectural Cancer Center, Ota 373-8550, Gunma, Japan; skoizuka@gunma-cc.jp

**Keywords:** naldemedine, opioid-induced constipation, performance status, real-world data, treatment patterns

## Abstract

*Background and Objectives*: Naldemedine is a peripherally acting μ-opioid receptor antagonist that improves opioid-induced constipation. Although clinical trials have excluded patients with poor performance status (PS) and those started on naldemedine early after opioid initiation, clinical practice has used naldemedine for the same patients. Therefore, we investigated the treatment patterns of naldemedine in a real-world setting. *Materials and Methods*: This was a multicenter, retrospective chart review study of opioid-treated patients with cancer receiving naldemedine. Adverse events that occurred within 7 days of naldemedine initiation were evaluated in those who received one or more doses of the same. Effectiveness was assessed in patients who used naldemedine for more than 7 days. *Results*: A total of 296 patients satisfied the eligibility criteria, among whom 129 (43.6%) had a PS of ≥3 and 176 (59.5%) started naldemedine within 2 weeks of opioid initiation. Moreover, 203 (79.6%) patients had ≥3 bowel movements per week. Incidences of all grades of diarrhea and abdominal pain were 87 (29.4%) and 12 (4.1%), respectively. No patient had grade 4 or higher adverse events. *Conclusions*: Although nearly half of the patients receiving naldemedine in clinical practice belonged to populations that were not included in the clinical trials, our results suggested that naldemedine in clinical practice had the same efficacy and safety as that in clinical trials.

## 1. Introduction

Opioid analgesics are used to relieve various severe pains, especially those associated with cancer. The analgesic effects of opioids are enforced by binding to various opioid receptors (μ, κ, and δ-opioid receptors) in the central nervous system. Morphine, oxycodone, and fentanyl, which activate the μ-opioid receptor, have mainly been used for pain management in patients with cancer. They can also activate μ-opioid receptors in the gastrointestinal tract, resulting in gut hypomotility, and opioid-induced constipation (OIC) has been one of the most common adverse effects. OIC, which can be characterized as functional constipation, has been defined as a change from baseline bowel habits and defecation patterns following the initiation of opioid therapy [1]. Reports have shown a 56% incidence rate of OIC in Japan when diagnosed according to Rome IV diagnostic criteria [2].

Naldemedine is a peripherally acting μ-opioid receptor antagonist (PAMORA) that improves symptoms in patients with OIC by binding to opioid receptors in the gastrointestinal tract [3]. The efficacy and safety of naldemedine in OIC have been confirmed through several clinical trials and meta-analyses [4,5,6,7,8,9,10]. Among these, COMPOSE-4 and COMPOSE-5 clinical trials have confirmed the efficacy and safety of naldemedine in patients with cancer [6,7]. However, these trials only included patients who had an Eastern Cooperative Oncology Group performance status (PS) ≤ 2 and were receiving a stable daily dose of opioids for 2 weeks before screening. In other words, the efficacy and safety of patients with a poor PS (PS ≥ 3) and those who started naldemedine early (≤2 weeks after opioid initiation) have yet to be investigated in clinical trials. Moreover, the National Comprehensive Cancer Network (NCCN) guidelines recommend that PAMORAs, such as naldemedine, be considered when response to laxative therapy is insufficient for OIC [11]. In clinical practice, however, naldemedine has often been used immediately after starting opioids or in combination with other laxatives.

The current study therefore sought to investigate a real-world clinical issue involving the actual use of naldemedine in patients who do not satisfy the eligibility criteria for clinical trials or guideline recommendations. To clarify the issue, we conducted a multicenter, retrospective study focusing particularly on the treatment patterns of naldemedine for patients with cancer using opioids in actual clinical practice as well as examining its efficacy and safety.

## 2. Materials and Methods

### 2.1. Study Population

This multicenter retrospective chart review study included patients with cancer receiving naldemedine at 10 institutes across Japan. This study was approved by the ethics committees of the participating institutions. The requirement for informed consent was waived owing to the retrospective nature of the study. However, the opportunity to refuse participation through an opt-out method was guaranteed. Patients prescribed naldemedine on admission between June 2017 and August 2019 were identified from electronic medical records and the pharmacy database. Those who satisfied the following criteria were then included for analysis: (1) pathologically diagnosed with a malignant tumor; (2) naldemedine initiated during hospitalization; (3) naldemedine used in combination with opioids; (4) availability of medical records on the treatment course spanning until 7 days after naldemedine treatment.

### 2.2. Clinical Data

Data on patient background, treatment, naldemedine effects, and adverse events were extracted from the electronic medical charts. Adverse events occurring within 7 days of naldemedine initiation were evaluated in patients who received one or more doses of the same. Adverse events were graded using the Common Terminology Criteria for Adverse Events version 5.0 (CTCAE v5.0).

Effectiveness was assessed in patients who used naldemedine for more than 7 days. The presence of three or more bowel movements during the first week of naldemedine was considered a clinically meaningful situation and was defined as a responder in this study. Patients with colostomies were excluded given the difficulty in counting the number of bowel movements.

### 2.3. Statistical Analysis

Data from each institution were anonymized, collected from the data managing office of the Division of Pharmacy, Gunma Prefectural Cancer Center, and then analyzed. The correlation between the incidence of diarrhea and duration of opioid use before naldemedine initiation was evaluated using the chi-squared test. All statistical analyses were performed using EZR (Saitama Medical Center, Jichi Medical University, Saitama, Japan), which is a graphical user interface for R (The R Foundation for Statistical Computing, Vienna, Austria) [12], with *p* values less than 0.05 indicating statistical significance.

## 3. Results

### 3.1. Patients

A total of 536 patients prescribed naldemedine during their hospitalization were identified, among whom 296 satisfied the eligibility criteria and were included in this study (Figure 1). Patient demographics and baseline clinical characteristics are listed in Table 1. The median age was 72 years (range: 33–96 years), with 118 (39.9%) patients aged 75 years or older and 129 (43.6%) patients with poor PS (PS ≥ 3). Thoracic cancer was the most common malignancy type in this cohort (27.7%), followed by pancreatic cancer (13.9%), and colorectal cancer (10.1%). The median regular opioid dose in oral morphine equivalents was 30 mg/day (range: 8–800 mg). Oxycodone was the most commonly used opioid, with 93 patients (32.4%) receiving 10 mg of oxycodone (15 mg of morphine equivalent) (Table 2). Moreover, 233 (78.7%) patients received concomitant laxatives, among whom 190 (81.5%) received magnesium oxide. Furthermore, 176 (59.5%) patients started naldemedine within 2 weeks of opioid initiation, with 36 patients (12.2%) notably starting both on the same day.

### 3.2. Efficacy and Safety

After excluding three patients with colostomies, the effectiveness of naldemedine was assessed in 255 patients who took naldemedine for more than 7 days. One patient had a colostomy and discontinued naldemedine after 5 days of treatment. As shown in Figure 2, 203 patients (79.6%) had at least three bowel movements per week. Incidences of adverse events likely caused by naldemedine are summarized in Table 3. Accordingly, 29.4% and 4.1% of the patients experienced diarrhea and abdominal pain (all grades) within the first 7 days of treatment, respectively. No patient had grade 4 or higher adverse events based on our evaluation using CTCAE v5.0. Patients with a shorter duration (≤14 days) of opioid use before starting naldemedine had a significantly lower incidence of diarrhea compared to those with longer use opioid use (≥14 days) (23.3% vs. 37.8%, *p* = 0.007; Figure 3).

## 4. Discussion

This has been the first study to investigate the treatment patterns of naldemedine in patients with cancer receiving opioids in a real-world setting. Notably, our finding showed that nearly half of the patients in actual clinical practice had poor PS or were started on naldemedine early after opioid initiation.

COMPOSE-4 and COMPOSE-5, randomized phase III studies on naldemedine in patients with cancer who exhibited OIC, excluded patients with PS ≥ 3 and those who started naldemedine within 2 weeks after opioid initiation [6]. On the other hand, 43.6% of the patients included in the current study had PS ≥ 3, whereas 59.5% were started on naldemedine within 2 weeks of opioid initiation. This suggests that more than half of the patients receiving naldemedine in real-world clinical practice did not belong to the population whose efficacy and safety have been confirmed in clinical trials. Although elderly patients may generally be excluded from clinical trials, COMPOSE-4 and -5 included patients over 20 years of age, with no upper age limit [6]. In a subgroup analysis of phase III trials, naldemedine had been reported to be generally well tolerated and effective in patients over 65 years of age suffering from chronic non-cancer pain [13]. This analysis appears to suggest that naldemedine can be effectively and safely used to treat older populations.

The NCCN guidelines recommend the use of PAMORAs, including naldemedine, when laxative therapy, such as magnesium oxide, could not address OIC [11]. In actual clinical practice, however, this recommendation seems to be rarely followed. Notably, nearly half of the patients (41.6%) included herein started naldemedine within 7 days of opioid initiation, with 12.2% of the patients starting both on the same day. One of the reasons why real-world utilization differs from that recommended in the guideline may be that the Japanese package insert (Symproic^®^) does not mention the timing of administration. Moreover, several studies have recently suggested that starting naldemedine early after opioid initiation can reduce the frequency of diarrhea [14,15,16]. Indeed, similar results had been obtained in the present study, with lower incidences of diarrhea having been observed when naldemedine was started within 14 days of opioid initiation. Therefore, the use of naldemedine in the early stages of opioid treatment, as practiced in real-world clinical settings, may become widespread in the near future. No studies have yet compared naldemedine to other laxatives, and naldemedine is not recommended as the first choice for laxative therapy [11]. The MAGNET study, a randomized controlled trial comparing naldemedine and magnesium oxide for OIC, is currently ongoing, the results from which will certainly be helpful in selecting the first-line drug for OIC [17].

Among the patients included herein, 79.3% can be characterized as responders, that is, those who had three or more bowel movements per week. Although the current study adopted a relatively loose definition given our lack of specification for an increase in the number of bowel movements prior to the start of naldemedine, our results were comparable to those presented in the COMPOSE-4 study (71%) [6]. While combining naldemedine with other laxatives may affect its efficacy, the combination rate observed herein (78.7%) was similar to that reported in the COMPOSE-4 study (74.2%).

In Japanese clinical trials on patients with cancer exhibiting OIC, the most common adverse reactions with naldemedine (0.2 mg) were diarrhea (19.6–39.7%) and abdominal pain (1.7%) [6,8], with the current study showing similar incidence rates of diarrhea (29.4%) and abdominal pain (4.1%). One mechanism for the development of diarrhea and abdominal pain with naldemedine is the inhibition of peripheral μ-receptors, which has been known to cause peripheral opioid withdrawal symptoms [18].

The current study has some limitations worth noting. First, given the retrospective nature of this study, the efficacy of naldemedine could not be evaluated based on “spontaneous” bowel movements, and data were limited to inpatients. Although several outpatients are available in actual clinical practice, the data of inpatients have been more reliable than those of outpatients given that they are evaluated from multiple perspectives by different healthcare providers. Another limitation is that the decision to start or discontinue naldemedine administration was left to the discretion of each physician and was not standardized. However, we believe that these limitations indicate that the results of this study reflect actual clinical practice and are well worth reporting. Moreover, data from the 296 cases included herein represent the largest cohort of real-world patients to our knowledge.

## 5. Conclusions

In conclusion, although nearly half of the patients in real-world settings did not satisfy the eligibility criteria for clinical trials, including those with poor PS or use of naldemedine early after opioid initiation, the efficacy and safety of naldemedine in real-world settings were still considered equivalent to those in clinical trials.

## Figures and Tables

**Figure 1 medicina-57-01233-f001:**
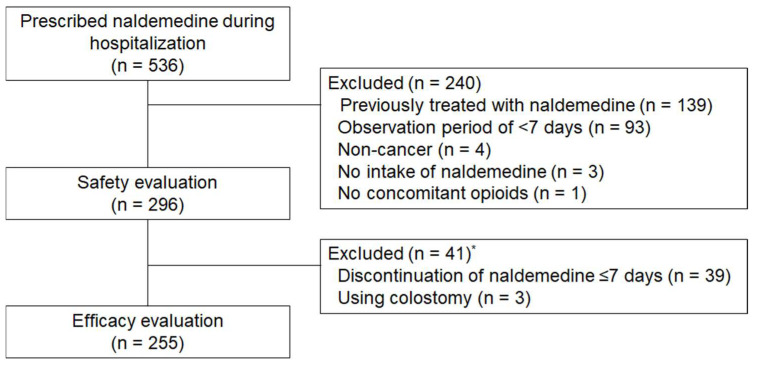
Flow diagram of patient enrollment. * One patient who had a colostomy discontinued naldemedine on day 5.

**Figure 2 medicina-57-01233-f002:**
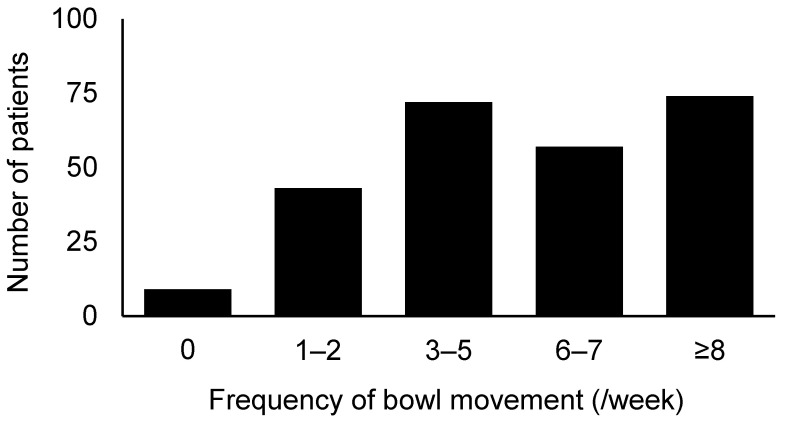
Frequency of bowel movements after naldemedine intake.

**Figure 3 medicina-57-01233-f003:**
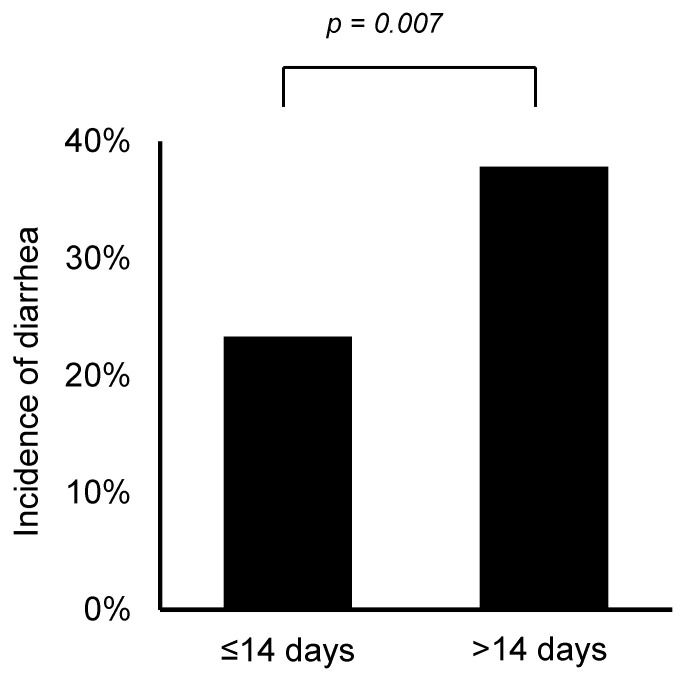
Incidence of diarrhea according to duration of opioid use before starting naldemedine.

**Table 1 medicina-57-01233-t001:** Patient demographic and baseline characteristics cited.

Parameter	*n* (%)
Male	179 (60.5%)
Age	
≤64	77 (26.0%)
65–74	101 (34.1%)
≥75	118 (39.9%)
ECOG PS	
0	24 (8.1%)
1	55 (18.6%)
2	88 (29.7%)
3	92 (31.1%)
4	37 (12.5%)
Primary tumor	
Thoracic	82 (27.7%)
Pancreatic	41 (13.9%)
Colorectal	30 (10.1%)
Gastrointestinal	22 (7.4%)
Breast	17 (5.7%)
Blood	14 (4.7%)
Prostate	11 (3.7%)
Gynecologic	11 (3.7%)
Head and neck	10 (3.4%)
Others	58 (19.6%)
Treatment *	
Anticancer medications	80 (27.0%)
Radiation	33 (11.1%)
Surgery	6 (2.0%)
Chemoradiation	4 (1.4%)
Past treatment	
Surgery (abdominal)	102 (34.5%)
Radiation (abdominal/lumbar)	35 (11.8%)
Comorbidities related to cancer	
Peritonitis	33 (11.1%)
Brain tumor/metastasis	25 (8.4%)
Meningitis	3 (1.0%)
Gastrointestinal obstruction	1 (0.3%)
Diabetes mellitus	
Yes	35 (11.7%)
No	261 (88.3%)

* Treatment of cancer in 3 weeks before starting naldemedine. ECOG PS, Eastern Cooperative Oncology Group performance status.

**Table 2 medicina-57-01233-t002:** Summary of opioid and laxative use.

Parameter	*n* (%)
Daily dose of opioids *	
≤19 mg	106 (36.9%)
20–49 mg	112 (39.0%)
50–99 mg	35 (12.2%)
≥100 mg	34 (11.8%)
Regular use of opioids *	
Oxycodone	165 (57.5%)
Morphine	49 (17.1%)
Fentanyl	42 (14.6%)
Hydromorphone	21 (7.3%)
Others	10 (3.5%)
Days from opioid initiation to starting naldemedine ^†^	
≤3 days	68 (23.0%)
4–7 days	55 (18.6%)
8–14 days	53 (17.9%)
15–29 days	46 (15.5%)
30–99 days	43 (14.5%)
≥100 days	30 (10.1%)
Coadministration of laxatives ^‡^	
Magnesium oxide	190 (81.5%)
Sennoside	81 (34.8%)
Bisacodyl	31 (13.3%)
Lubiprostone	26 (11.2%)
Sodium picosulfate hydrate	23 (9.9%)
Others	20 (8.6%)

* Oral morphine equivalent to regular opioids. Nine patients who did not use regular opioids were excluded when calculating the percentages. ^†^ The number of days of use in one patient who had started opioids at another hospital could not be determined. ^‡^ Percentages were calculated for 233 patients who were using laxatives concomitantly.

**Table 3 medicina-57-01233-t003:** Adverse events considered to be associated with naldemedine use.

	Grade 1	Grade 2	Grade 3	Grade 4
Diarrhea	61 (20.6%)	20 (6.8%)	6 (2%)	0 (0.0%)
Abdominal pain	7 (2.4%)	4 (1.4%)	1 (0.3%)	―
Nausea	17 (5.7%)	8 (2.7%)	3 (1%)	―
Anorexia	30 (10.1%)	9 (3%)	4 (1.4%)	0 (0.0%)
Vomiting	3 (1%)	3 (1%)	2 (0.7%)	0 (0.0%)
Fatigue	19 (6.4%)	4 (1.4%)	3 (1%)	―

## Data Availability

The data presented in this study are available on request from the corresponding author. The data are not publicly available.

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
