# Peer review of "Real-World Patient Characteristics and Treatment Patterns of Naldemedine for the Treatment of Opioid-Induced Constipation in Patients with Cancer: A Multicenter Retrospective Chart Review Study"

_medicina, 2021, doi:10.3390/medicina57111233_

Round 1
Reviewer 1 Report
After reading the manuscript entitled “Real-word treatment patterns of naldemedine for patients with cancer receiving opioids: A multicenter retrospective chart review study”, I consider that the topic of the paper does not have the clinical relevance.
The intention of the study is good, since it is important to know what happens in clinical practice beyond what is reflected in the written recommendations. And as the authors say, although the National Comprehensive Cancer Network (NCCN) guidelines recommend that µ-opioid receptor antagonist (PAMORA), such as naldemedine, be considered when response to laxative therapy is insufficient for opioid induced constipation. In clinical practice, however, naldemedine has often been used immediately after starting opioid or in combination with other laxatives.
So the results obtained by this work are predictable and not considered relevant for a post.
Reviewer 2 Report
medicina-1387385, Real-world treatment patterns of naldemedine for patients with cancer receiving opioids: A multicenter retrospective chart review study, presents a small, but interesting and important research. The paper is fit for Medicina journal’s scope.
The title should better reflect the content of the paper.
The authors should present and comment similar work and highlight the originality of their data in this context.
See for example:
Efficacy of naldemedine for the treatment of opioid-induced constipation: A meta-analysis, J Gastrointestin Liver Dis. 2019 Mar;28(1):41-46
Safety and Efficacy of Naldemedine for the Treatment of Opioid-Induced Constipation in Patients with Chronic Non-Cancer Pain Receiving Opioid Therapy: A Subgroup Analysis of Patients ≥ 65 Years of Age, Drugs Aging. 2020 Apr;37(4):271-279. doi: 10.1007/s40266-020-00753-2
There are some minor problems that need to be corrected.
Row 56, the authors should discuss more on the fact that some opioid analgesics are in fact antagonist on miu-receptors, but agonists on ĸ opioid receptor.
Row 65, “improves OIC” could be changed to “improved symptoms in patients with OIC”
Most tables are just descriptive data. The authors should try to add more statistical analyses results.
The editing of the paper should be corrected in some sections to correspond with that of the journal.
Figure 2. For me a pie graph is not the best way to presents those data. Also, the quality of the image could be improved to be more attractive visually.
